# Potential Hormetic Effects of Cimetidine on Aerobic Composting of Human Feces from Rural China

**Xiaowei Li [1,\*], Xuan Wang [1], Xusheng Pan [1], Ping Zhu [1,\*], Qianzhi Zhang [2], Xiang Huang [1], Xiuquan Deng [3], Zhipu Wang [4], Yao Ding [1], Ximing Liu [1] and John L. Zhou [5]**

1 School of Environmental and Chemical Engineering, Organic Compound Pollution Control Engineering, Ministry of Education, Shanghai University, Shanghai 200444, China
2 Instrumental Analysis & Research Center, Sun Yat-sen University, Guangzhou 510275, China
3 Guangxi Liyuanbao Science and Technology Co., Ltd., Nanning 530000, China
4 State Key Laboratory of Heavy Oil Processing, China University of Petroleum-Beijing at Karamay, Karamay 834000, China
5 School of Civil and Environmental Engineering, University of Technology Sydney, Sydney, NSW 2007, Australia
\* Correspondence: lixiaowei419@shu.edu.cn (X.L.); xhnzp@shu.edu.cn (P.Z.); Tel.: +86-021-66137773 (X.L.)

**Abstract:** Aerobic composting is widely used worldwide as a natural process for handling human waste. Such waste often contains pharmaceutical residues from human consumption, yet their impact on composting has not been studied. The aim of this study is to investigate the impact of the antihistamine cimetidine (10 mg/kg, 100 mg/kg) on the aerobic composting of human feces. The key results show that 10 mg/kg of cimetidine accelerates temperature increase and moisture removal of the composting substrate. The organic matter in all the groups gradually decreased, and the pH values increased first and then declined with the composting time, with no significant differences between the groups. The $NH_4^+$-N concentrations and $NH_3$ emission reached the maximum at 1.5 days and then declined rapidly, while the $NO_2^-$-N concentrations increased and then decreased, and the $NO_3^-$-N contents tended to increase all the time during the composting. The 100 mg/kg cimetidine caused a higher maximal $NH_4^+$-N concentration of compost, and a lower maximal $NH_3$ emission at 1.5 days, while 10 mg/kg cimetidine led to more $NO_2^-$-N and $NO_3^-$-N contents. In addition, 10 mg/kg cimetidine enhanced the aromatization and humification of dissolved organic matter and promoted the degradation of aliphatic substances. Furthermore, 100 mg/kg cimetidine generated a larger influence on the microorganisms than 10 mg/kg cimetidine, especially for the microorganisms related to nitrogen transformation. The findings imply that cimetidine has a dose-dependent impact on the decomposition of organic matter and the conversion of nitrogen in human feces during composting. It deserves further investigation of the possible hormesis effect.

**Keywords:** human; aerobic composting; cimetidine; potential effect; nitrogen conversion

## 1. Introduction

Pharmaceuticals and personal care products (PPCPs) are extensively applied for health care and animal husbandries such as antibiotics, antihistamines, and cytostatic drugs, and in people's daily lives, such as sunscreens, antibacterial agents, preservatives, and household detergents [1]. These compounds, representing a wide range of human consumption products, are discharged into the environment because of widespread consumption, limited human metabolism, and inappropriate management [2]. The global demand for medicine is increasing with the impact of the 2019 coronavirus disease [3], and the scale is expected to reach USD 1.06 trillion in 2024 [4]. Once these pharmaceuticals enter the environment, their negative effect is generated on untargeted organisms, even at trace concentrations. Jechalke et al. reported that several PPCPs, such as antibiotics, can decrease soil nitrification and respiration rates and generate an effect soil microbial community [5]. Besides the

impacts on the soil ecosystem, PPCPs can also interfere with the human endocrine system. More than one-third of pesticides and their metabolites exhibit human antiglucocorticoidic activity and have cumulative or synergistic effects of endocrine disruptors [6]. Furthermore, these transformation products of pesticides may increase hormone secretion and associated gene expression levels [7].

Cimetidine is a widely-used typical H2 antihistamine for treating stomach illness [8]. Large amounts of cimetidine are consumed every year [9]. The annual consumption of cimetidine was almost 160 tons in the USA [10] and about 125–130 tons in Korea [11]. A high content of cimetidine is emitted into the environment since pharmaceuticals such as cimetidine cannot be removed effectively by conventional wastewater treatment [12]. The residual medicine will be released into the soil through sewage recycling and sewage sludge application as fertilizer. Pérez-Carrera et al. reported that the concentration of cimetidine is the most abundant (7.8 ng/g) of all the tested medicines in the sediment near a wastewater treatment plant in Spain [13]. A low concentration of cimetidine is found in soil (<1.69 ng/g dw), but the plants can enrich cimetidine up to 997 ng/g dw, according to the Dynamic Plant Uptake model [14]. Some studies investigated the potential risks of cimetidine residues to ecosystems and human receptors. Lee et al. reported that cimetidine could disrupt the endocrine system after short-term exposure and change the steroidogenic pathway and sex hormone balance of adult zebrafish, such as the development of gonadal intersex in the female fish [15]. Hoppe et al. found that long-term exposure to cimetidine negatively affects aquatic invertebrate growth and population dynamics [16]. Manzato et al. suggested that cimetidine also causes androgenic failure in the serum testosterone levels of rodents [17]. In fact, humans do not fully absorb cimetidine when taking this drug, and sixty percent of cimetidine is excreted in the form of human excreta [18], causing high contents of cimetidine in human waste. At present, it is unclear about the effect of cimetidine on the treatment and disposal of human excreta.

Co-composting of crop residues and feces in rural areas is an efficient treatment method to reduce the pollution of solid waste and promote the reuse of the waste [19]. Composting can efficiently eliminate persistent organic compounds such as PPCPs, and pathogenic microorganisms in human feces [20]. In turn, the organic compounds in human feces may exert an influence on the composting process. Antibiotics are capable of affecting the release of ammonia, causing the pH decline during composting process [21]. Nitrogen is an important macronutrient for cell growth and biochemical processes of the composting process [22]. The five main stages of the nitrogen cycle—ammonification, nitrification, denitrification, anammox, and assimilation—are all largely controlled by microbial activity [23]. However, the pollutants of raw materials can influence the nitrogen transformation during composting. Vieuble Gonod et al. found that sulfonamides inhibit the N cycle process and enhance the concentration of total nitrogen content (TN) during composting due to the hormesis effect [24]. A high concentration of amoxicillin leads to lower $NO_2^-$-N and $NO_3^-$-N contents in the thermophilic period, implying that the presence of amoxicillin weakens the nitrification [25]. They may result from the high sensitivity of the nitrogen-converted microorganism to high contents of medicine [26]. The addition of oxytetracycline changes the nitrogen transformation by reducing nitrifying bacteria and the emergence of *Bacillus* and *Thermobifida* [27].

Humification is a nutrient-stabilizing process of compost, which is also affected by medicine. Penicillin G reduces the synthesis of humus and humic acid by disrupting the metabolisms of amino acids and carbohydrates in the microorganism communities [28]. Composting is directly tied to microbial succession since the organic matter degradation is mostly carried out by a variety of microorganisms [29]. The abundance of medicine also exerts an influence on the microbial population. Guo et al. (2022) found a decrease in Firmicutes and an increase in *Bacteroidetes* at the end of the composting in Chinese herbal medicine [30]. The studies imply that various PPCPs, such as pharmaceuticals, have complex and unpredictable effects on the nitrogen conversion, humification, and microorganism community in aerobic composting. Therefore, it is hypothesized that

cimetidine has a considerable effect on composting due to the potentially high content of cimetidine in human feces.

The objects of the study are to (1) investigate the impact of the cimetidine on the treatment performance of the human-feces composting; (2) explore the evolution of dissolved organic matters (DOMs) using fluorescence and Fourier transform infrared (FTIR) spectroscopy; (3) analyze the microbial community succession during the composting through high-throughput sequencing technologies. The study will give a detailed understanding of the potential effects of cimetidine on the treatment and disposal of human feces, particularly for composting process.

## 2. Materials and Methods

### 2.1. Samples and Composting Reactor

The human-feces samples were taken from a rural village in Wuwei County, Wuhu City, Anhui Province, China. The collected samples were then stored in plastic containers at 4 °C until the experiment began. Rice bran (1–2 mm) as bulking material of composting was gained in a food supplier in Panjin City, Liaoning Province, due to its highly porous structure and specific surface area [31]. Table S1 displays the physicochemical properties of human feces and rice bran.

Three 10-L bench-scale compost reactors were set up, as shown in Figure S1. Each reactor was covered by a thermal insulator to maintain temperature and reduce heat loss during composting. The perforated bottom plate of the reactors was used for the aeration process. The aeration rate was 0.25 L (air)/(min·kg) with 40 min on/20 min off during the human-feces composting [32].

### 2.2. Human Feces Composting

A total of 6.3 kg of human feces and 4.5 kg of rice bran (wet weight) were fully mixed. The carbon-nitrogen ratio (C/N) and moisture content of the mixtures were around 16 and 60% [33]. Then, the substrates (10.8 kg in total) were divided into three groups, with an average of 3.6 kg per group. Two cimetidine concentrations (10 and 100 mg/kg) were prepared in the initial mixtures based on the daily consumption dosage (www.drugbank.ca/drugs/DB005, acessed on 14 February 2019, the excreted percentage of the medicine [16], and the excrement mass per capita [34]. 10 mg/kg is the average daily dosage of cimetidine, and 100 mg/kg is the higher concentration to investigate the effect on the environment. The three groups are shown as followings: two experiment groups, composting with 10 mg/kg cimetidine, composting with 100 mg/kg cimetidine; and one control group, composting without cimetidine. Deionized water without cimetidine was added to the mixture as the control group. Then the mixtures were added to the reactors for a 21-day composting [25]. Approximately 50 g of the samples were collected from the reactor's upper, middle, and bottom sections per 1.5 days during days 1–3 and per 3 days during days 3–21. Two subsamples were taken from each sample. One subsample was kept at −20 °C for analyzing the physicochemical characteristics of the substrates, while the other was kept at −80 °C for the analysis of microbial community using 16S rDNA high-throughput sequencing.

### 2.3. Compost Physicochemical Properties

The temperatures were measured at the reactors' top and bottom sections with a K-type thermometer every 12 h. The moisture content was estimated by drying the samples in a 105 °C oven for 24 h. The dried composting samples were then burned at 550 °C for four hours in a muffle furnace to determine the content of volatile solids (VS). The pH value of the samples was determined using pH meters in deionized water with a solid-to-water ratio of 1:10 [35]. $NH_3$ content was measured following the Chinese HJ 533-2009 National Standard, using a 0.01 mol/L $H_2SO_4$ solution for titration. $NH_4^+$-N, $NO_2^-$-N, and $NO_3$-N contents of the samples were determined according to the reference (APHA, 2005). Each test was carried out in triplicate.

### 2.4. Characterization of Dissolved Organic Matters

DOMs were extracted from the samples according to the reference [36]. A Multi N/C 2100 TOC analyzer (Analytikjena, Jena, Germany) was used to estimate the soluble TOC contents (STOC) of the samples. Fluorescence spectroscopy was used to analyze the fluorescent components in the DOM samples obtained at 1.5, 6, and 21 days based on the reference [37], using an F-7000 fluorescence spectrophotometer (Hitachi, Tokyo, Japan). Three-dimensional(3D) excitation/emission matrix (EEM) spectra were acquired by scanning the excitation wavelength between 200 to 500 nm and the emission wavelength between 250 to 600 nm. The EEM spectral data were analyzed using Fluorescence Regional Integration (FRI) techniques [37]. Convectional emissions and excitation spectra of the DOM samples were also gained with a speed of 12,000 nm/min based on the reference [38]. In addition, the DOM samples collected at 0, 1.5, 3, 6, 9, 12, 15, and 21 days were measured using Fourier transform infrared (FTIR) spectroscopy of Nicolet 389 spectrometer (Thermo Fisher, Tokyo, Japan) according to the reference [39].

### 2.5. Microbial Community Analysis

The microbial community of the samples collected at 1.5, 6, and 21 days were analyzed using 16S rRNA high-throughput sequencing analysis after their DNA was extracted using the E.Z.N.A.® Soil DNA Kit (Omega Bio-tek, Norcross, GA, U.S.) based on the instruction according to our previous study [36]. A NanoDrop2000 UV-vis spectrophotometer was used to measure the content and purity of the DNA samples. Illumina MiSeq sequencing of the DNA samples was carried out by Majorbio-Biopharm Biotechnology Co., Ltd. in Shanghai, China. The original sequencing quality was controlled by Trimmomatic (Illumina, California, U.S.) and FLASH (U.S.) software (Version 1.2.11). UPARSE software (Version 7.1) was used to analyze operational taxonomic units (OTUs) with a similarity level of 97%. Each sequence labeled using the RDP classifier was further compared with the Silva database (SSU123) (Bremen, Germany) with an alignment threshold of 70%. The Heatmap and cluster analysis were performed with Origin 2022 to present the species composition and abundance information among different samples at the genus level. The differences in the distribution of microorganisms among various samples were assessed through principal component analysis (PCA) using Majorbio official website tools.

## 3. Results and Discussion

### 3.1. Effect of Cimetidine on Treatment Performance of Human Feces Composting

3.1.1. Physicochemical Properties of the Bulk Substrate

Temperature variation reflects microbial activity and progress relating to the mineralization and humification of composting [34]. The temperatures in all the groups rapidly increased and then decreased with the composting time (Figure 1a). The control, 10 mg/kg, and 100 mg/kg groups arrived at the maximum temperature at 0.5–1.0 days, which is 40.4, 42.4, and 39.4 °C, respectively. As shown in Figure 1b, the cumulative temperature increase in the 10 mg/kg group is much higher than that of the other two groups. The results indicate that the presence of low-dose cimetidine promotes the enhancement of aerobic composting temperature due to the higher microbial activity [40]. They correspond to the previous study about the effect of ranitidine, possibly due to the hormesis effect [41].

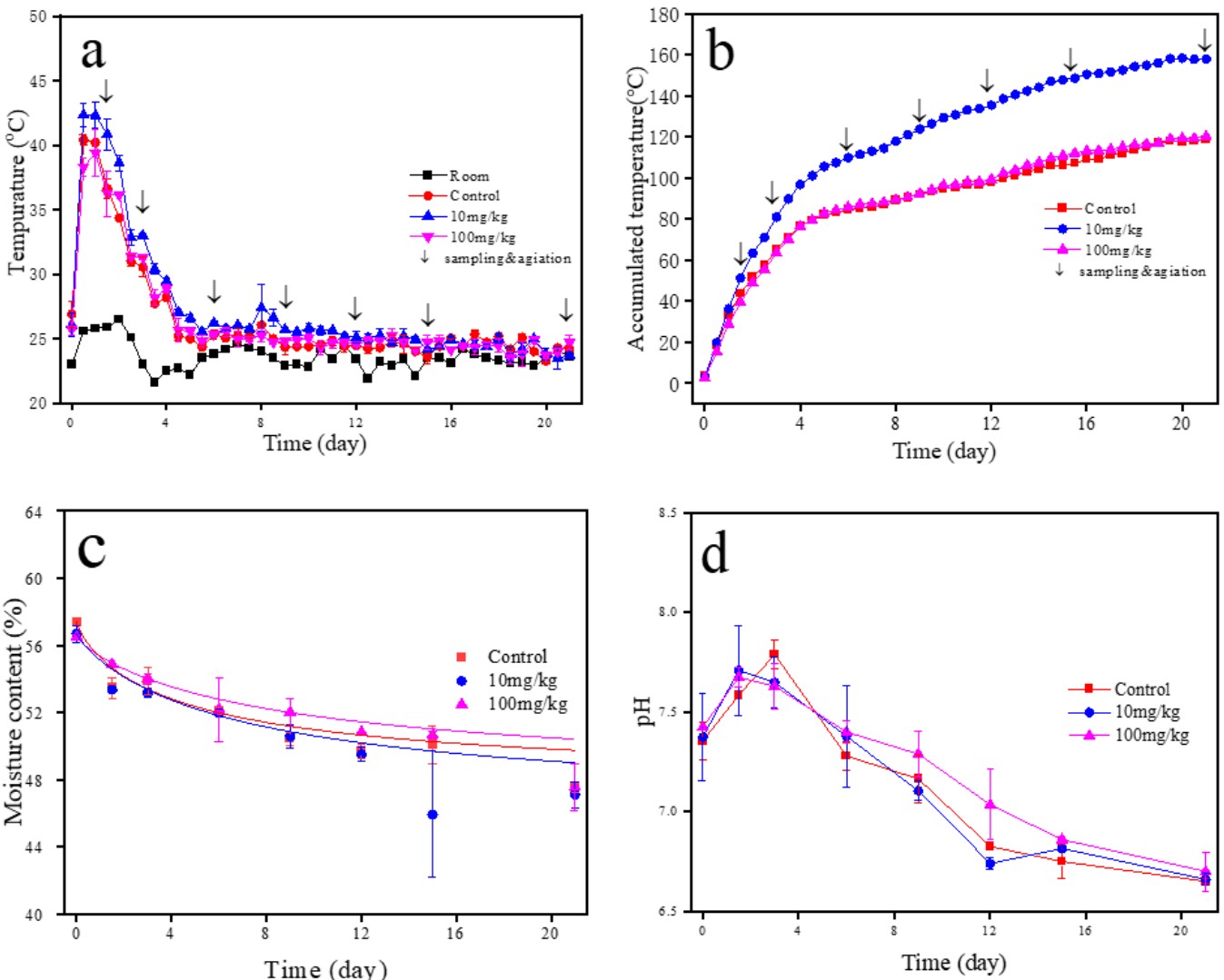

**Figure 1.** Changes in temperature (**a**), accumulated temperature (**b**), moisture content (**c**), and pH value (**d**) in the control, 10 mg/kg, and 100 mg/kg groups as the composting of human feces.

The moisture content is linked to the dissolved organic matter and enzyme activity, thus affecting the microbial succession [42]. Figure 1c shows that the moisture contents in the initial thermophilic period of the groups rapidly decrease with the composting time, implying the high microorganism activity because of the high-temperature [32]. After 21 days, the 10 mg/kg group's moisture content was lower than the control and 100 mg/kg group, in accordance with the temperature results (Figure 1b), implying that the 10 mg/kg group had more water removal. Figure 1d shows that pH values present a similar variation in the groups; all display the first rise and then a decline during the composting process. The pH increase possibly resulted from the deterioration of organic acids and the formation of ammonium nitrogen ($NH_4^+$-N) [21]. In contrast, the pH decrease may be ascribed to the following hydrogen ions accumulation caused by the nitrification [43].

### 3.1.2. Transformation of Carbon-Containing Materials

Human feces possess various organic matter, such as sugars, proteins, fats, and cellulose [44], which can supply the carbon or energy sources for microorganisms' growth and metabolism. The VS contents of all the groups gradually decreased over the operation time (Figure S2), indicating organic matter degradation, and stabilization of the composting substrate [43]. As shown in Figure 2, STOCs of all the groups decreased very quickly in six days,

implying that the labile organic matter is degraded rapidly in the initial stage of aerobic composting [45]. Then, the STOC contents tended to stabilize over 6–21 days, corresponding to the results of Xu et al. [18]. At the end of composting, the STOC contents decreased to 0.80 mg/g, 0.89 mg/g, and 0.92 mg/g for the control, 10 mg/kg, and 100 mg/kg groups, respectively. Overall, the groups had the similar changes in the VS and STOC contents, indicating that cimetidine had no undetectable influence on the decomposition of organic matter during the human feces composting.

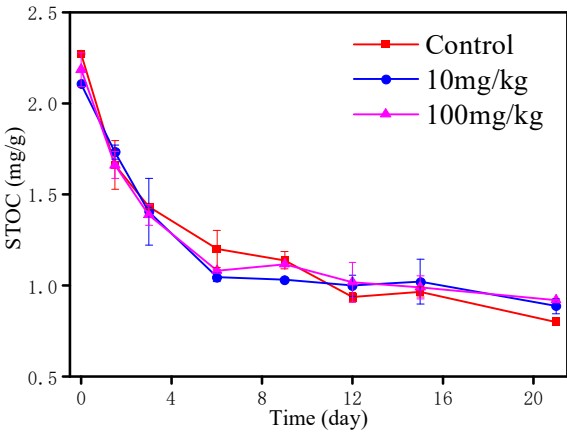

**Figure 2.** Changes in STOC contents in the control, 10 mg/kg and 100 mg/kg groups as the composting of human feces.

### 3.1.3. Transformation of Nitrogen-Containing Materials

Nitrogen is a critical component of microbial cell structure materials, an essential component for microbial activity and a source of synthetic nitrogen-containing metabolites during composting [25]. Figure 3a shows that the $NH_4^+$-N concentration in the 100 mg/kg group reached the maximum values (4.72 mg/kg) at 1.5 days, which is higher than the 10 mg/kg group (2.70 mg/kg), followed by the control group (1.89 mg/kg). The results indicate that the cimetidine promotes the ammoniated process, especially for the 100 mg/kg group. Then, the $NH_4^+$-N concentrations tended to decrease, possibly due to the nitrification by ammonia oxidizing microorganisms and nitrite oxidizing bacteria (NOB) [25]. $NH_4^+$-N content in each group was below the threshold (0.40 g/kg) after 21 days, suggesting the compost maturity [46]. As shown in Figure 3b, the $NH_3$ emissions in all the groups reached the maximum also at 1.5 days (Figure 3b). The 100 mg/kg group had much lower $NH_3$ emission, compared with the two other groups, inferring that the high dosage of cimetidine inhibited the $NH_3$ emission during the composting. The sequential decrease in $NH_4^+$-N content and $NH_3$ emission after 1.5 days was possibly associated with the decreasing degradation of nitrogen-containing organic matter as well as the nitrification of $NH_4^+$-N [47]. The $NO_2^-$-N concentrations in all the groups were very low in the first 3 days, then firstly increase in 3–12 days and decrease after 12 days (Figure 3c). The low $NO_2^-$-N contents for the first 3 days may be attributed to high sensitivity of nitrifying bacteria to high temperature in the initial composting, leading to inhibition of the nitrification [48]. The reduction of $NO_2^-$-N concentration after 12 days resulted from the oxidation of $NO_2^-$-N by the NOB [49]. Meanwhile, the $NO_3^-$-N contents in all the groups fluctuate in first 9 days, and then rapidly increase in 9–21 days (Figure 3d). Generally, the 10 mg/kg group has the highest $NO_2^-$-N content but the lowest $NO_3^-$-N contents after 9 days in all the groups. They imply that the 10 mg/kg of cimetidine may exert a great adverse effect on oxidation of $NO_2^-$-N by NOB. Therefore, the results demonstrate that the presence of cimetidine may boost the ammoniation process but inhibit the nitrite-oxidizing process during aerobic composting.

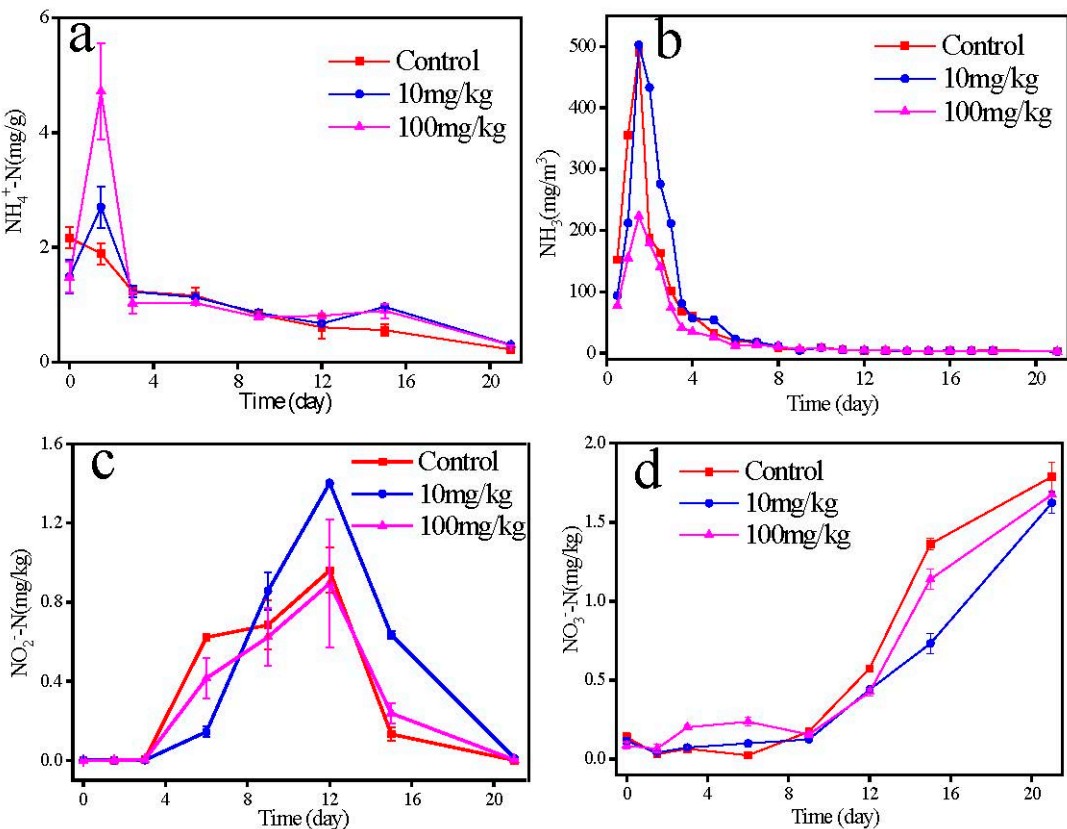

**Figure 3.** Changes in $NH_4^+$-N (**a**), $NH_3$ (**b**), $NO_2^-$-N (**c**), and $NO_3^-$-N contents (**d**) in the control, 10 mg/kg, and 100 mg/kg groups as the composting of human feces.

### 3.2. DOMs Analysis Using Fluorescence and FITR Spectroscopy

DOMs represent the most active component of the compost organic matter [50], which not only contain a large number of biological by-products but can also supply a direct nutrition source for microorganism growth. The quality of composting products is often evaluated through the chemical composition and structure of the DOMs [51]. Therefore, it is significant to investigate the effect of cimetidine on the DOMs during the human feces composting process.

#### 3.2.1. Fluorescence Spectroscopy

Fluorescence spectroscopy is often used to measure the changes in organic fluorescent groups, e.g., proteins, humic acids, and fulvic acids, during biological treatments [52]. The EEM spectra of DOMs at 1.5, 6, and 21 days are shown in Figure 4. Four characteristic peaks are found in the EEM spectra. Peak 1 (Ex/Em = 225–230 nm/310–325 nm) and Peak 2 (Ex/Em = 275–280 nm/320–350 nm) relate to the soluble microbial by-product and aromatic proteins such as tyrosine-like and tryptophan-like groups [38], while Peak 3 (Ex/Em = 325–340 nm/420–430 nm) and Peak 4 (Ex/Em = 285–290 nm/415–425 nm) represent the compounds such as humic acid and fulvic acid [53].

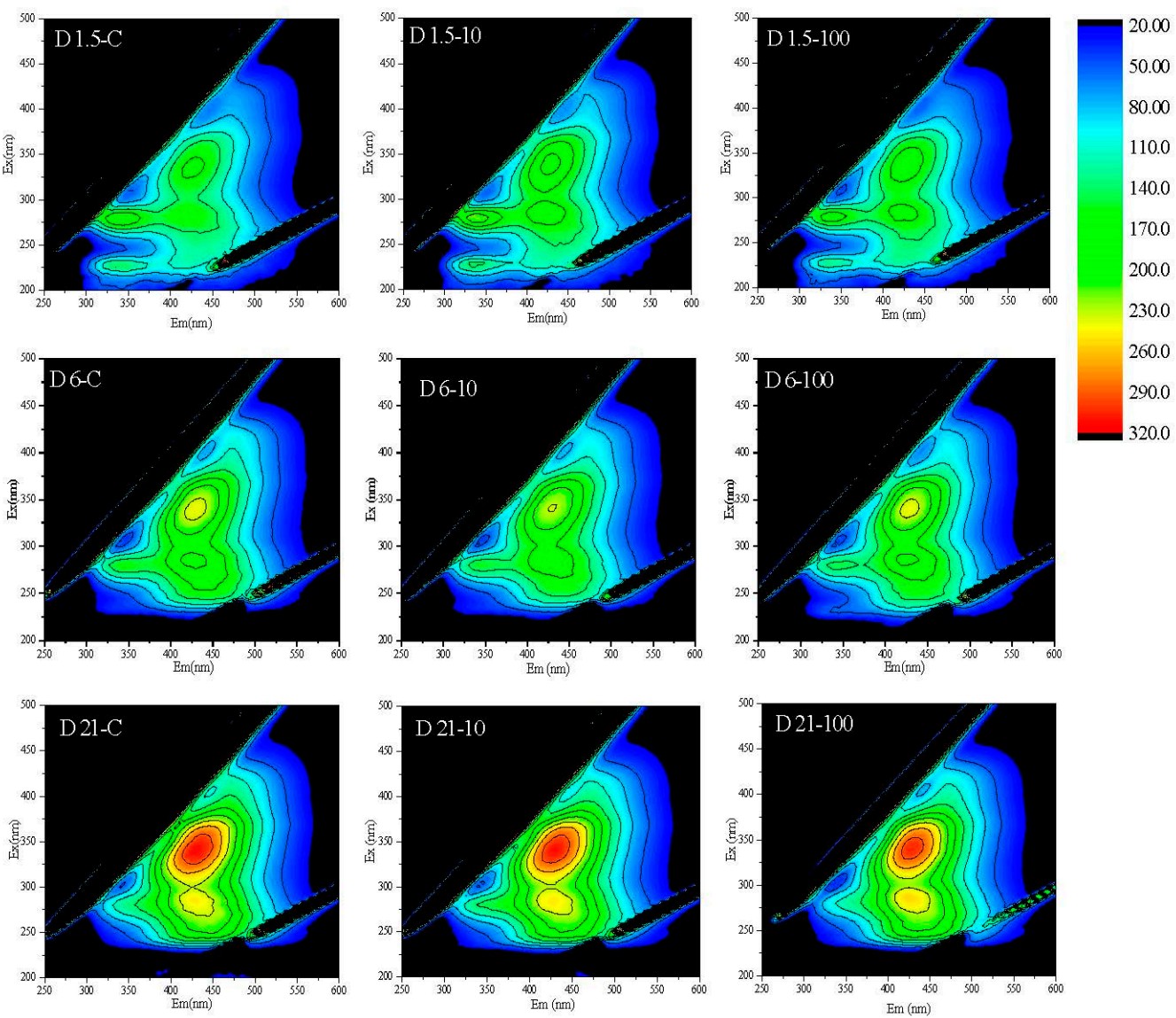

**Figure 4.** Fluorescence EEM spectra of DOMs at 1.5, 6, and 21 days (D) in the control (C), 10 mg/kg (10), and 100 mg/kg (100) groups as the composting of human feces.

Specific fluorescence intensities (SFIs) of Peaks 1 and 2 tended to decrease. At the same time, those of Peaks 3 and 4 had an increasing tendency as the operation time, indicating the decomposition of soluble microbial by-product and protein-like groups, the accumulation of humic acid and fulvic acid during the composting of human excrement [54]. Microorganisms use degradable and unstable DOMs, such as proteins and soluble microbial by-products, for metabolic activities and generate more stable compost with humus macromolecular organic matter during the composting [55]. SFIs of all four peaks in the 10.mg/kg group were greater than that in the others at 1.5 days (Table 1), implying that the 10 mg/kg group has higher organic fluorescent matter. They are attributed to higher microbial activity causing the transformation of more organic matter to fluorescent DOMs due to higher temperature in the 10 mg/kg group (Figure 1).

**Table 1.** Ex/Em maxima and specific fluorescence intensity (SFI) of EEM spectra of human feces aerobic composts. DOMs at 1.5, 6, and 21 days.

| Samples | Peak 1 | | Peak 2 | | Peak 3 | | Peak 4 | |
|---|---|---|---|---|---|---|---|---|
| | Ex/Em | SFI | Ex/Em | SFI | Ex/Em | SFI | Ex/Em | SFI |
| 1.5 days | | | | | | | | |
| Control | 225/320 | 110.35 | 280/325 | 174.81 | 335/425 | 178.93 | 285/420 | 167.71 |
| 10 mg/kg | 225/325 | 130.79 | 280/325 | 194.71 | 340/430 | 209.83 | 285/425 | 192.40 |
| 100 mg/kg | 225/325 | 115.59 | 280/325 | 169.41 | 340/430 | 191.33 | 285/420 | 182.51 |
| 6 days | | | | | | | | |
| Control | 230/315 | 19.82 | 280/325 | 135.91 | 340/430 | 240.13 | 285/425 | 210.10 |
| 10 mg/kg | 230/325 | 27.73 | 280/325 | 128.11 | 340/430 | 232.83 | 285/425 | 193.90 |
| 100 mg/kg | 230/325 | 45.17 | 280/325 | 128.91 | 340/430 | 240.73 | 285/425 | 208.30 |
| 21 days | | | | | | | | |
| Control | 230/325 | 9.00 | 280/325 | 96.85 | 335/425 | 311.01 | 285/425 | 247.55 |
| 10 mg/kg | 230/325 | 15.62 | 280/325 | 116.15 | 340/430 | 312.41 | 285/425 | 247.15 |
| 100 mg/kg | 230/325 | 10.57 | 280/325 | 90.05 | 340/430 | 296.81 | 285/425 | 241.45 |

The FRI technique is a quantitative method, integrating the defined EEM region according to the fluorescence intensity [56], which can be used to analyze the fluorescence intensity data and overcome the overlapping and heterogeneity of four products' EEM spectra. $\Phi_i$ denotes the accumulated fluorescence intensity of organics with similar characteristics by normalizing $\Phi_i$ to relative regional areas to obtain the Ex/Em area volumes ($\Phi_{(i,n)}$, $\Phi_{(T,n)}$) and percent fluorescent response ($P_{(i,n)}$). The $P_{(1,n)}$, $P_{(2,n)}$, and $P_{(3,n)}$ in all the groups show a decreasing tendency with the composting time, and the $P_{(4,n)}$ and $P_{(5,n)}$ tended to increase (Figure S3), implying a decrease in the percentages of protein-like groups and an increase in the percentages of humic acid-like materials during the composting. Compared with the other groups, higher $\Phi_{(T,n)}$ in the 10 mg/kg group implies higher fluorescence organic matter in the samples. The possible reason is that low-dosage cimetidine enhances the organic matter degradation, and the enrichment of fluorescence organic matters such as aromatic protein and humic acid-like groups, in accordance with the previous studies based on ranitidine [36].

Studies have shown that the concentration of humic acid-like groups positively correlates with the fluorescence intensities of emission spectra in the 380–550 nm range with a fixed excitation wavelength of 360 nm and excitation spectra at a fixed emission wavelength of 520 nm in the range of 300–500 nm [57]. The intensities of the emission and excitation spectra from the three groups gradually increase with the composting time, indicating the enhancement of the humic acid-like groups (Figure S4). The 10 mg/kg group has higher intensities of emission and excitation spectra compared with the other two groups, further confirming that the DOMs in the 10 mg/kg group have a greater degree of humification and stability [57].

### 3.2.2. FTIR Spectroscopy

As shown in Figure S5, the main changes in the FTIR spectra are outlined with composting time as followings: (1) a decrease in the peak intensity at 2930 cm$^{-1}$ and 2855 cm$^{-1}$ relating to C–H stretching of aliphatic substances [58]; (2) an intensity increase at 1630–1640 cm$^{-1}$ (C=O stretch of the amide groups and C=C of aromatic groups) [59]; (3) a weakening of peak intensity at 1546–1553 cm$^{-1}$ (N-H stretching of amide II) [54]; (4) an intensity decrease at 1050–1080 cm$^{-1}$ (C-O bond of polysaccharides) [51]. They indicate the loss of aliphatic, protein, and polysaccharides substances and the enrichment of aromatic compounds during the composting. The 10 mg/kg and 100 mg/kg groups exhibit greater reduction at about 2930 and 2855 cm$^{-1}$ than the control group and even disappear after 21 days, implying that the cimetidine accelerates the degradation of aliphatic substances in composting of human feces.

### 3.3. Effect of Cimetidine on Microbial Community Succession during Human Feces Composting

Microbial community in the three groups at 1.5, 6, and 21 days of composting are shown in Figure 5 at the phylum level. The top 10 bacteria accounted for about 99% of the total number of microbes, namely Proteobacteria (37.20–70.53%), Firmicutes (19.68–47.7%), Actinobacteria (1.79–13.31%), Bacteroidetes (1.91–9.70%), Patescibacteria (0.14–2.32%), Chloroflexi (0.33–1.35%), Myxococcota (0.04–1.03%), Verrucomicrobiota (0–0.71%), and Planctomycetes (0–1.73%). Awasthi et al. reported that Proteobacteria, Actinobacteria, and Bacteroidetes play vital roles in organic matter degradation during composting [55]. Firmicutes are the dominant bacteria in the mesophilic and thermophilic periods of the composting process and can degrade lignin and cellulose [25]. The relative abundance of Actinobacteria gradually increased as the composting proceeded, implying the enhancement of compost decay [60]. The 10 mg/kg group had a high relative abundance of Proteobacteria compared with the control and the 100 mg/kg groups at 6 and 21 days. They indicate that the growth of Proteobacteria may promote at low dosage of cimetidine, but inhibit at high dosage of cimetidine, possibly due to the hormesis effect.

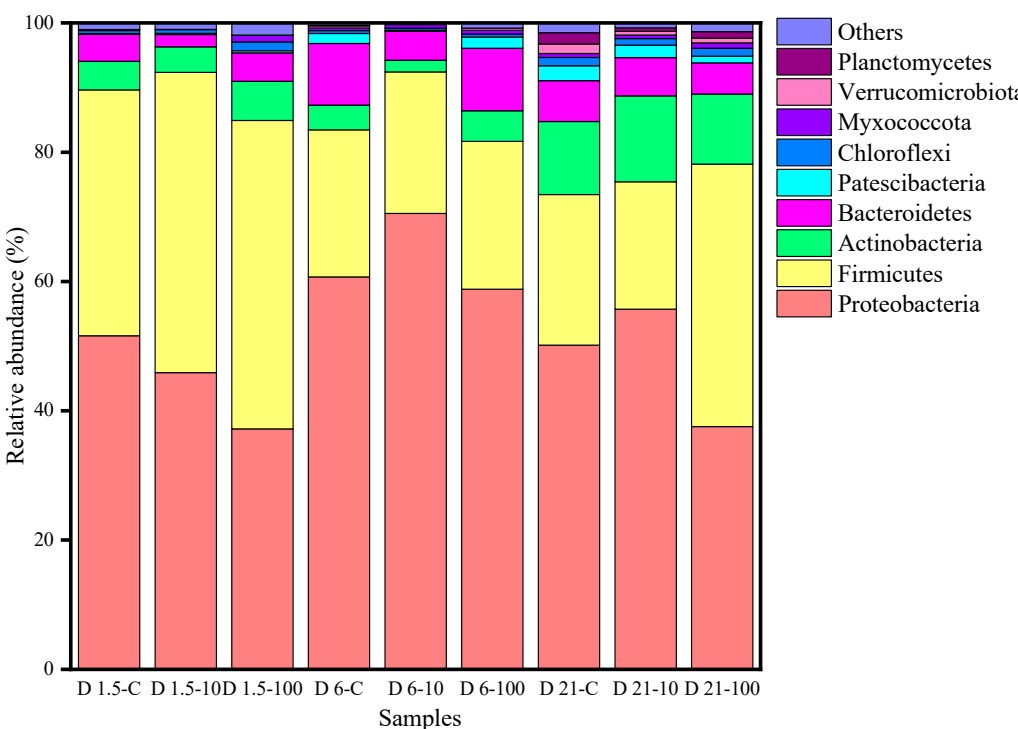

**Figure 5.** Changes in phylum-level microbial composition at 1.5, 6 and 21 days (D) in the control (C), 10 mg/kg (10) and 100 mg/kg (100) groups.

As shown in Figure S6, a heat map of top 20 abundant bacteria among the samples was constructed at the genus level. *Clostridium sensu stricto* was the dominant bacteria in the composting substrates, corresponding to the result by Chen et al. [52]. They can degrade lignocellulose and polysaccharides to form acetates and butyrate [61]. The 10 mg/kg group had higher abundance of *Clostridium sensu stricto* than the control and 100 mg/kg groups at 1.5 days, corresponding to the above result of temperature, possibly due to the hormesis effect [41]. *Parapusillimonas* abundance in the control (2.75%) and 10 mg/kg (3.50%) groups was higher than the 100 mg/kg group (1.88%) at 1.5 days. *Parapusillimonas* is associated with heterotrophic nitrification and denitrification of compost and wastewater [62], implying that a high dosage of cimetidine may inhibit the nitrification bacteria, thus causing the $NH_4$-N accumulation (Figure 3a). At 6 days, 10 mg/kg group had lower abundance of *Hydrogenophaga* than the control and 100 mg/kg groups, corresponding to the above result of $NO_2^-$-N concentration. At 21 days, the control group had the highest abundance

of *Ochrobactrum* in all the three groups, in accordance with the $NO_3^-$-N results. The results indicate that the cimetidine may exert a differentiate effect on the microorganisms relating to the nitrite and nitrate nitrogen conversion during the human feces compositing [63–66]. The previous study has also found that cimetidine had significant effects on respiration of heterotrophic microbial biofilms, and antihistamines can significantly alter bacterial community composition, resulted in significant relative increases in *Pseudomonas* sp. and decreases in *Flavobacterium* sp. [67]. As shown in Figure 6, the cluster analysis show that 10 mg/kg group had a more similar microorganism composition to the control group, compared with 100 mg/kg group for all the samples, implying that the high-dosage cimetidine may have higher influence on the microorganisms than the low-dosage cimetidine. Meanwhile, principal component analysis also reveals that the samples may be separated into three groups representing the various periods of composting, according to the microbial composition. This result demonstrates that the composting period generates an essential influence on bacterial community formation, similar to previous studies [68].

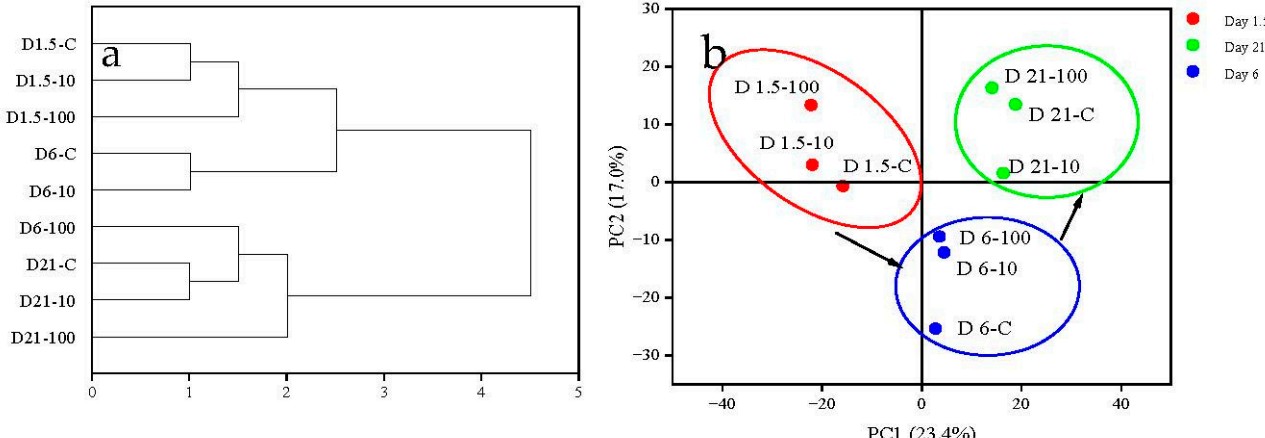

**Figure 6.** Cluster analysis (**a**) of the samples at 1.5, 6, and 21 days (D) in the control (C), 10 mg/kg (10) and 100 mg/kg (100) groups based on microbial composition genus level and principal component analysis (**b**) of the samples at 1.5, 6, and 21 days (D) in the control (C), 10 mg/kg (10) and 100 mg/kg (100) groups based on microbial composition OTU level.

## 4. Conclusions

The water removal, organic matter degradation, $NH_4$-N reduction, and increased degree of humification of human feces substrate happen after the aerobic composting, implying that composting enhances the maturity and stability of human feces. Cimetidine promotes an increase in the temperature and a reduction of the moisture content of the compost. The possible reason is that it promotes the degradation of aliphatic substances and the humification and aromatization of organic matter. Meanwhile, the presence of cimetidine may boost the ammoniation process of organic nitrogen but inhibit the nitrite-oxidizing process during aerobic composting. In addition, cimetidine enhances the relative abundance of Proteobacteria and *Clostridium sensu stricto*, and a high dosage of cimetidine has a greater effect on the microorganism composition. In sum, cimetidine has a dose-dependent influence on the organic matter and nitrogen conversion during human feces composting, possibly due to the hormesis effect.

**Supplementary Materials:** The following supporting information can be downloaded at: https://www.mdpi.com/article/10.3390/su142114454/s1. Table S1 Physicochemical properties of initial composting materials. Figure S1. Schematic diagram of an aerobic composting reactor. Figure S2. VS changes in the control, 10-mg/kg and 100-mg/kg groups during human feces composting. Figure S3. Normalized excitation-emission area volumes (a) and percent fluorescence response (b) of DOMs isolated from the composts. Figure S4. Emission (left) and Excitation (right) spectra of DOMs extracted from human feces aerobic composts. Figure S5. Changes in the FTIR spectra of DOMs

during human feces aerobic composting. Figure S6. Heat map of top 20 abundant bacteria at the genus level in the composting samples.

**Author Contributions:** Conceptualization, X.L. (Xiaowei Li) and P.Z.; methodology, X.W. and X.P.; software, X.W. and X.P.; validation, X.P. and X.H.; investigation, X.P., Y.D., and X.L. (Ximing Liu); resources, X.D.; data curation, X.W. and X.H.; writing—original draft preparation, X.W. and X.P.; writing—review and editing, X.L. (Xiaowei Li), P.Z., J.L.Z., Q.Z., and Z.W.; visualization, X.W. and X.P.; supervision, X.L. (Xiaowei Li) and P.Z.; project administration, X.L. (Ximing Liu) and X.D.; funding acquisition, X.L. (Xiaowei Li) and P.Z. All authors have read and agreed to the published version of the manuscript.

**Funding:** This research was funded by the National Key R&D Program of China (2018YFC1903202 and 2018YFC1903201), the National Natural Scientific Foundation of China (52070126), and the Shanghai Committee of Science and Technology (22WZ2505300 and 19DZ1204702).

**Institutional Review Board Statement:** Not applicable.

**Informed Consent Statement:** Not applicable.

**Data Availability Statement:** Not applicable.

**Conflicts of Interest:** The authors declare no conflict of interest.

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
