# Peer review of "Potential Hormetic Effects of Cimetidine on Aerobic Composting of Human Feces from Rural China"

_sustainability, doi:10.3390/su142114454_

Round 1

Reviewer 1 Report

The manuscript entitled "Potential hormetic effects of cimetidine on aerobic composting of human feces from rural China" investigate the impact of antihistamine cimetidine (10 mg/kg, 100 mg/kg) on aerobic composting of human feces. The topic is meaningful. But in my opinion the manuscript is not suitable for publication in the present form and require revisions indicated below.

Specific comments:

1. The form of manuscript is weird, and please revise it as the requirements of this journal.

2. Table 1: part of content is missing.

3. Figure 1a: the temperature of composting does not meet the basic criterion.

4. Fluorescence EEM spectra of DOMs in figure 4 needs to be improved about the quality of figure.

5. Figure 6b: the sequencing repeats should be given as the form of plot in PCA.

6. The reason of two cimetidine concentrations (10 and 100 mg/kg) in the initial mixtures is better to give more explanitions.

Author Response

Please see the attachment, thanks for your comments.

Reviewer 2 Report

The authors evaluation the effects of cimetidine on composting of fecal matter. A number of analytical tests are done, but there is no connection of the effects of this pharmaceutical to work done. Cimetidine is a common antihistamine. The effects of this drug on the immune system of animals is well characterized, but the effect on prokaryotes are not described in the article. Why should this drug have any effect on the experiment? This is basic information that should explain why the experiments were done and why differences between control and experimental treatments might exist.

FRI was done to evaluate levels of microbial waste product, aromatic amino acids, and humic/fulvic acids. How specific are these assays for the 4 products? 

A percent composition of microbes was calculated. I believe the authors also need to calculate the titer of microbes during this experiment. I would expect the titer to increase over time.  This would allow the authors to better correlate the abundance of denitrifying bacteria with ammonia levels in the compost. The weakness of this paper is not being able to link these two together. I think more definitive experiments need to be done to determine if this link is real.

The authors have not taken the time to define the abbreviations used in this paper. See PPCP, TN, etc. This makes reviewing and reading this paper difficult.

Line 175.  Two cimetidine concentration were used. These correspond to 3 biological levels of this compound. Please make these correlate so readers can understand.

A description of the N cycle should go into the introduction.

Author Response

Thanks for your comments, please see the attachment.

Reviewer 3 Report

The article is investigating the influence of the residual pharmaceutical compounds (antihistamine cimetidine) arising from human consumption, on the composting process of human feces. The findings imply that cimetidine has a dose-dependent impact on the decomposition of organic matter and the conversion of nitrogen in human feces during composting and observe the possible hormesis effect.

L155-156 What is the volume and general specifications of the human feces samples.

L166-167 (The rate was 0.25 L/(min·kg) with 40 minutes 166 on/20 minutes off during the human-feces composting (Guo et al.). Please clarify: what rate is it? Is it the rate of aeration? And is the volume unit (L) measuring? Is it the air or the liquid.

L272-273  (The pH increase is possibly resulted from the deterioration of organic acids and the formation of ammonium nitrogen). Remove the word (is).

L169. The section (2.2. Human feces composting) should be explained more clearly. A Table can be added to demonstrate the three mentioned group. It is not clear if every group is composed of different sample.

L172-173 (Then, the substrates was divided into three groups with an aver-173 age of 3.6 kg per group.). First change was to were and still the sentence is unclear.

Effect on treatment performance of human feces composting

L245 (3.1. Effect on treatment performance of human feces composting).The title is not clear. Normal it should follow the pattern (effect of xx on yy)

L815-816 (Figure 1. Change in temperature (a), accumulate temperature (b), moisture content (c) and pH value 815 (d) in the control, 10-mg/kg, and 100-mg/kg groups as the composting of human feces.

Th figure caption is not sufficiently illustrative and the objective of the analysis is not clear.

Correct accumulate temperature to accumulated temperature.

All figure captions are lacking clarity and objective.

The article concludes that composting enhances the maturity and stability of human feces, while cimetidine promotes in the temperature and  reduces the moisture content of the compost through the degradation of aliphatic substances, and the humification and aromatization of organic matter. But these suggested mechanisms should be fully explained in the discussion. The article does not show the significance and application of the obtained results.

The English language should be revised by a Native speaking expert for grammar and scientific styling.

Author Response

(The authors gave the same response as above.)

Round 2

Reviewer 1 Report

There are a lot of data for this paper, which has improved after revision. However, I still have some little questions as follows:

1. L358 The conclusion is advised not to describe something for further study.

2. The duration time for high-temperature stage may be not enough for safe composting.

3. There is little degradation of VS and loss of moisure. So I am doubt whether the composting is successful.

Reviewer 2 Report

With one exception, the authors addressed my comments nicely in their response for review. Unfortunately, most of this text was not inserted into the ms. Please remedy this.

The lone exception was the effects of cimetidine on microbial and protist propagation. This lead me to do a search of the literature on this topic. I suggest reading the following paper and inserting some of this information into your ms: Ecological Applications, 23, 2013, pp. 583–593. I think these data and findings support your arguments.

Reviewer 3 Report

The authors have responded sufficiently and correctly to the comments
